# Plasmon–emitter interactions at the nanoscale

P.A.D. Gonçalves [1,2,3,4]*, Thomas Christensen [1]*, Nicholas Rivera [1], Antti-Pekka Jauho [3,5], N. Asger Mortensen [3,4,6]* & Marin Soljačić [1]

Plasmon–emitter interactions are of central importance in modern nanoplasmonics and are generally maximal at short emitter–surface separations. However, when the separation falls below 10–20 nm, the classical theory deteriorates progressively due to its neglect of quantum effects such as nonlocality, electronic spill-out, and Landau damping. Here we show how this neglect can be remedied in a unified theoretical treatment of mesoscopic electrodynamics incorporating Feibelman $d$-parameters. Our approach incorporates nonclassical resonance shifts and surface-enabled Landau damping—a nonlocal damping effect—which have a dramatic impact on the amplitude and spectral distribution of plasmon–emitter interactions. We consider a broad array of plasmon–emitter interactions ranging from dipolar and multipolar spontaneous emission enhancement, to plasmon-assisted energy transfer and enhancement of two-photon transitions. The formalism gives a complete account of both plasmons and plasmon–emitter interactions at the nanoscale, constituting a simple yet rigorous platform to include nonclassical effects in plasmon-enabled nanophotonic phenomena.

[1] Department of Physics, Massachusetts Institute of Technology, Cambridge, MA 02139, USA. [2] Department of Photonics Engineering, Technical University of Denmark, DK-2800 Kgs., Lyngby, Denmark. [3] Center for Nanostructured Graphene, Technical University of Denmark, DK-2800 Kgs., Lyngby, Denmark. [4] Center for Nano Optics, University of Southern Denmark, Campusvej 55, DK-5230 Odense M, Denmark. [5] Department of Physics, Technical University of Denmark, DK-2800 Kgs., Lyngby, Denmark. [6] Danish Institute for Advanced Study, University of Southern Denmark, Campusvej 55, DK-5230 Odense M, Denmark. *email: pa@mci.sdu.dk; tchr@mit.edu; asger@mailaps.org

The interaction between light and matter in free-space is an intrinsically weak process. Strikingly, the interaction strength can be enormously enhanced near material interfaces. This is especially true in plasmonics[1–6]: for an emitter separated from an interface by a subwavelength distance $h$, the decay rate is increased by a factor $\propto h^{-3}$ in the classical, macroscopic theory. However, as the separation—or the characteristic dimension of the plasmonic system itself—is reduced to the nanoscale ($\lesssim 10–20$ nm), the classical theory is rendered invalid due to its neglect of all intrinsic quantum mechanical length scales in the plasmonic material. Thus, to ascertain the ultimate limits of plasmon-mediated light–matter interactions, the classical theory must be augmented.

In principle, time-dependent density functional theory (TDDFT)[7] may be used to describe plasmon excitations in a quantum mechanical setting. Unfortunately, its application imparts very substantial demands on the associated computational cost, effectively restricting applications of TDDFT in plasmonics to few-atom clusters[8–11] or highly symmetric few-nanometer-scale systems[12–15]. In fact, the vast majority of plasmonic designs—particularly those of relevance for enhancing light–matter interactions, where it is often the separation and not the system itself that is nanoscopic—fall outside this space. As such, neither a classical (macroscopic) nor a purely quantum mechanical (microscopic) approach can satisfactorily treat light–matter interactions in the multiscale yet nanoscopic systems of experimental relevance.

To overcome this, a mesoscopic treatment of light–matter interactions in nanoplasmonics can be developed whose applicability encompasses a wide range of length scales, and, in particular, bridges the gap between microscopic and macroscopic descriptions (Fig. 1). This framework, which is based on the so-called Feibelman $d$-parameters[16,17], facilitates a simultaneous incorporation of electronic spill-out, nonlocality, and surface-assisted Landau damping—all intrinsically quantum mechanical mechanisms—through a simple modification of the macroscopic framework, thereby enabling the calculation of plasmon-mediated light–matter interactions in the mesoscopic regime.

Here we report the impact of nonclassical corrections in a broad range of prominent plasmon-mediated light–matter interaction phenomena, namely, the Purcell—or, equivalently, local density of states (LDOS)—enhancement[18–21], the enhancement of dipole-forbidden (i.e., multipolar) transitions[22–24], plasmon-mediated energy transfer between two emitters[25–27], and finally the enhancement of two-photon processes for an emitter near a metal surface[28–30]. In all cases, we find substantial deviations from classicality when the emitter–metal separation or the intrinsic geometric parameters, like a sphere's radius, fall below $\sim 10$ nm. We identify two key mechanisms that produce these deviations: (i) surface-enhanced Landau damping, which broadens the plasmonic response; and (ii) nonclassical frequency shifts, toward the red in jellium and blue in noble metals. Intriguingly, these deviations become nonnegligible well-before a completely non-retarded regime is reached, demonstrating the existence of a multiscale competition between retardation and nonclassical corrections even at mesoscopic scales.

## Results

**Nonclassical optical response.** The optical response of any structure is encoded by a set of scattering coefficients: e.g., for a planar system, they are the reflection coefficients $\{r^{\mathrm{TM}}, r^{\mathrm{TE}}\}$—whose mesoscopic generalizations Feibelman introduced[16]—and for a spherical system, they are the Mie coefficients $\{a_l^{\mathrm{TM}}, a_l^{\mathrm{TE}}\}$—whose mesoscopic generalization we introduce here. These coefficients constitute the fundamental building blocks from which the optical response to all external stimuli—and associated

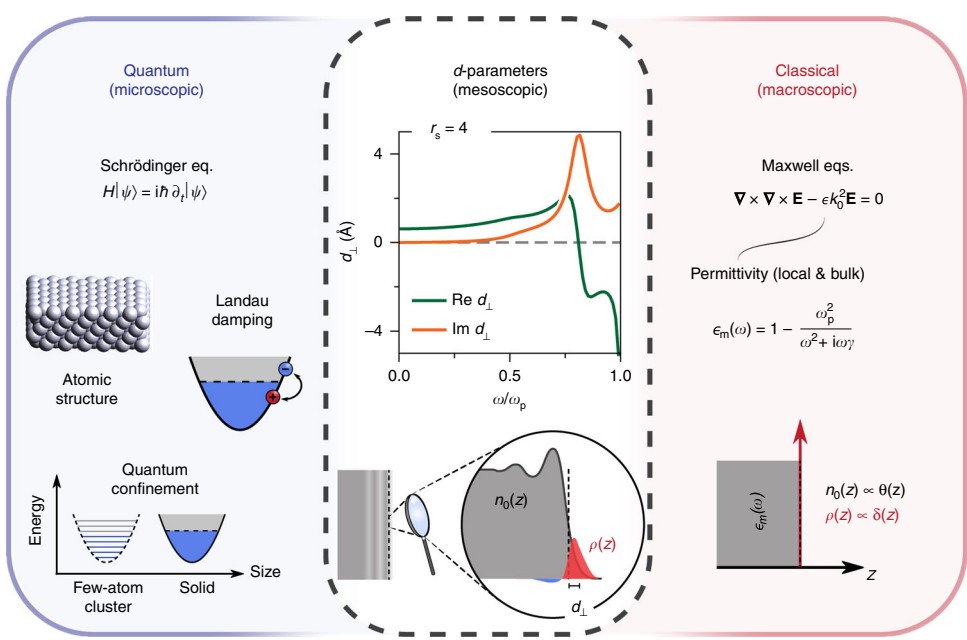

**Fig. 1 Nonclassical mesoscopic electrodynamics via $d$-parameters.** The nonclassical surface-response functions—the Feibelman $d$-parameters—rigorously incorporate quantum mechanical effects in mesoscopic electrodynamics, bridging the gap between the purely quantum (microscopic) and classical (macroscopic) domains. Inset: $d_\perp$-parameter of an $r_s = 4$ jellium computed from TDDFT[17]; the corresponding $d_\parallel$-parameter vanishes (due to charge-neutrality[33]).

nanophotonic phenomena such as plasmonic enhancements of light–matter interactions—can be inferred.

$d$-parameters through a generalization of the usual electromagnetic boundary conditions[36] (Supplementary Note 5). For a metallic sphere of radius $R$, we find that the generalized, nonclassical TM and TE Mie coefficients are

$$a_l^{\text{TM}} = \frac{\epsilon_{\text{m}} j_l(x_{\text{m}}) \Psi_l'(x_{\text{d}}) - \epsilon_{\text{d}} j_l(x_{\text{d}}) \Psi_l'(x_{\text{m}}) + (\epsilon_{\text{m}} - \epsilon_{\text{d}}) \{ j_l(x_{\text{d}}) j_l(x_{\text{m}})[l(l+1)] d_\perp + \Psi_l'(x_{\text{d}}) \Psi_l'(x_{\text{m}}) d_\parallel \} R^{-1}}{\epsilon_{\text{m}} j_l(x_{\text{m}}) \xi_l'(x_{\text{d}}) - \epsilon_{\text{d}} h_l^{(1)}(x_{\text{d}}) \Psi_l'(x_{\text{m}}) + (\epsilon_{\text{m}} - \epsilon_{\text{d}}) \{ h_l^{(1)}(x_{\text{d}}) j_l(x_{\text{m}})[l(l+1)] d_\perp + \xi_l'(x_{\text{d}}) \Psi_l'(x_{\text{m}}) d_\parallel \} R^{-1}}, \tag{4a}$$

$$a_l^{\text{TE}} = \frac{j_l(x_{\text{m}}) \Psi_l'(x_{\text{d}}) - j_l(x_{\text{d}}) \Psi_l'(x_{\text{m}}) + (x_{\text{m}}^2 - x_{\text{d}}^2) j_l(x_{\text{d}}) j_l(x_{\text{m}}) d_\parallel R^{-1}}{j_l(x_{\text{m}}) \xi_l'(x_{\text{d}}) - h_l^{(1)}(x_{\text{d}}) \Psi_l'(x_{\text{m}}) + (x_{\text{m}}^2 - x_{\text{d}}^2) h_l^{(1)}(x_{\text{d}}) j_l(x_{\text{m}}) d_\parallel R^{-1}}, \tag{4b}$$

For concreteness, we take a jellium metal with a Wigner–Seitz radius of $r_s = 4$ ($\hbar\omega_{\text{p}} \approx 5.9\,\text{eV}$; representative of Na[31,32]) throughout our calculations: the associated $d$-parameters are shown in Fig. 1. The corresponding classical response, $\epsilon_{\text{m}}(\omega) = 1 - \omega_{\text{p}}^2/(\omega^2 + i\omega\gamma)$, is of the Drude-type and we assume a damping rate corresponding to $\hbar\gamma = 0.1\,\text{eV}$. Additional results for the cases of an $r_s = 2$ jellium (representative of Al[31,32]) and Ag are given in Supplementary Notes 4 and 5. For a succinct description of the $d$-parameter formalism, see Methods (and Supplementary Note 1).

We first consider the simplest case, that of a planar dielectric–metal interface onto which a transverse magnetic (TM) or a transverse electric (TE) polarized plane-wave impinges from the dielectric side. The mesoscopic, Feibelman-$d$-parameter-corrected generalizations of the associated Fresnel reflection coefficients $r^{\text{TM}}$ and $r^{\text{TE}}$ are (Supplementary Note 3)[16,33,34]

$$r^{\text{TM}} = \frac{\epsilon_{\text{m}} k_{z,\text{d}} - \epsilon_{\text{d}} k_{z,\text{m}} + (\epsilon_{\text{m}} - \epsilon_{\text{d}})(iq^2 d_\perp - i k_{z,\text{d}} k_{z,\text{m}} d_\parallel)}{\epsilon_{\text{m}} k_{z,\text{d}} + \epsilon_{\text{d}} k_{z,\text{m}} - (\epsilon_{\text{m}} - \epsilon_{\text{d}})(iq^2 d_\perp + i k_{z,\text{d}} k_{z,\text{m}} d_\parallel)}, \tag{1a}$$

$$r^{\text{TE}} = \frac{k_{z,\text{d}} - k_{z,\text{m}} + (\epsilon_{\text{m}} - \epsilon_{\text{d}}) i k_0^2 d_\parallel}{k_{z,\text{d}} + k_{z,\text{m}} - (\epsilon_{\text{m}} - \epsilon_{\text{d}}) i k_0^2 d_\parallel}, \tag{1b}$$

with in-plane, free-space, and bulk wavevectors $q$, $k_0 \equiv \omega/c$, $k_j \equiv \sqrt{\epsilon_j} k_0$, respectively, and where $k_{z,j} \equiv \sqrt{k_j^2 - q^2}$ (for $j \in \{d, m\}$). Here, $\epsilon_{\text{d}} \equiv \epsilon_{\text{d}}(\omega)$ and $\epsilon_{\text{m}} \equiv \epsilon_{\text{m}}(\omega)$ denote the local bulk permittivities of the dielectric and metallic media, respectively. Importantly, all quantum mechanical contributions in Eqs. (1a) and (1b) are completely captured by the microscopic surface response functions $d_\perp$ and $d_\parallel$; the classical limit is naturally recovered when $d_{\perp,\parallel} \to 0$. The retarded surface plasmon-polariton (SPP) dispersion can be determined from the poles of the associated reflection coefficient for TM polarized waves [cf. Eq. (1a)], and thus follows from the solution of the implicit equation:

$$\frac{\epsilon_{\text{d}}}{\kappa_{\text{d}}} + \frac{\epsilon_{\text{m}}}{\kappa_{\text{m}}} - (\epsilon_{\text{m}} - \epsilon_{\text{d}}) \left( \frac{q^2}{\kappa_{\text{m}} \kappa_{\text{d}}} d_\perp - d_\parallel \right) = 0, \tag{2}$$

where $\kappa_j \equiv \sqrt{q^2 - k_j^2} = -i k_{z,j}$. In the nonretarded limit (where $\kappa_{\text{d,m}} \to q$), this reduces to the simpler condition:

$$\epsilon_{\text{m}} + \epsilon_{\text{d}} - (\epsilon_{\text{m}} - \epsilon_{\text{d}}) q (d_\perp - d_\parallel) = 0. \tag{3}$$

Naturally, the well-known retarded and nonretarded classical plasmon conditions, $\epsilon_{\text{d}}/\kappa_{\text{d}} + \epsilon_{\text{m}}/\kappa_{\text{m}} = 0$ and $\epsilon_{\text{m}} = -\epsilon_{\text{d}}$, respectively, are recovered in the limit of vanishing $d$-parameters.

While the mesoscopic reflection coefficients of the planar system were determined by Feibelman[16], the corresponding scattering coefficients of the spherically symmetric system—the so-called Mie coefficients $a_l^{\text{TM}}$ and $a_l^{\text{TE}}$ ref. [35]—have remained unknown, despite their significant practical utility. Here, we derive the mesoscopic generalization of Mie's theory by incorporating the Feibelman

with dimensionless wavevectors $x_j \equiv k_j R$, spherical Bessel and Hankel functions of the first kind $j_l(x)$ and $h_l^{(1)}(x)$, and the Riccati–Bessel functions $\Psi_l(x) \equiv x j_l(x)$ and $\xi_l(x) \equiv x h_l^{(1)}(x)$; primed functions denote their derivatives. Equations (4a) and (4b) constitute the spherical counterparts to the reflection coefficients of the planar interface. Like them, they directly determine the response of the scattering object, here a metallic sphere, to any external perturbation (in a basis of spherical vector waves; see Supplementary Note 5). For instance, the extinction cross-section is simply $\sigma_{\text{ext}} = 2\pi k_{\text{d}}^{-2} \sum_{l=1}^{\infty} (2l+1) \text{Re}(a_l^{\text{TM}} + a_l^{\text{TE}})$ ref. [35] with resonances determined by the poles of the nonclassical Mie coefficients. For subwavelength metal spheres, the optical response is primarily embodied in $a_l^{\text{TM}}$, which has a series of peaks that correspond to the excitation of localized surface plasmons (LSPs) of dipole, quadrupole, etc. character (for $l \in \{1, 2, \dots\}$, respectively)[35,37]. In the small-radius limit, $x_j \ll 1$, a small-argument expansion of spherical Bessel and Hankel functions produces the nonretarded equivalent of the TM Mie coefficient, the mesoscopic multipolar polarizability[38] (Supplementary Note 5)

$$\alpha_l = 4\pi R^{2l+1} \frac{(\epsilon_{\text{m}} - \epsilon_{\text{d}}) \left[ 1 + \frac{l}{R} \left( d_\perp + \frac{l+1}{l} d_\parallel \right) \right]}{\epsilon_{\text{m}} + \frac{l+1}{l} \epsilon_{\text{d}} - (\epsilon_{\text{m}} - \epsilon_{\text{d}}) \frac{l+1}{R} (d_\perp - d_\parallel)}. \tag{5}$$

In the nonretarded limit, the extinction cross-section $\sigma_{\text{ext}} \equiv \sigma_{\text{abs}} + \sigma_{\text{sca}}$ is dominated by $l = 1$ dipole contributions so that $\sigma_{\text{abs}} \simeq k_{\text{d}} \text{Im} \, \alpha_1$ and $\sigma_{\text{sca}} \simeq k_{\text{d}}^4 |\alpha_1|^2 / 6\pi$, peaking around the dipole LSP frequency[35]. More generally, the $l$th nonretarded LSP condition is set by the poles of $\alpha_l$:

$$\epsilon_{\text{m}} + \frac{l+1}{l} \epsilon_{\text{d}} - (\epsilon_{\text{m}} - \epsilon_{\text{d}}) \frac{l+1}{R} (d_\perp - d_\parallel) = 0. \tag{6}$$

Once again, Eqs. (4)–(6) reduce to their well-known classical counterparts when $d_{\perp,\parallel} \to 0$. It is interesting to note that the incorporation of quantum mechanical effects breaks the scale-invariance that usually characterizes the nonretarded classical limit, wherein plasmon resonances $\omega^{\text{cl}}$ are scale-independent (e.g., $\omega^{\text{cl}} = \omega_{\text{p}}/\sqrt{1 + \epsilon_{\text{d}}}$ and $\omega^{\text{cl}} = \omega_{\text{p}}/\sqrt{1 + 2\epsilon_{\text{d}}}$ for the surface and dipole plasmon of a planar and spherical jellium interface, respectively). Here, the introduction of the length scale(s) associated with $d_{\perp,\parallel}$ breaks this scale-invariance, producing finite-size corrections parameterized by either $q d_{\perp,\parallel}$ or $d_{\perp,\parallel}/R$, cf. Eqs. (3) and (6).

In this context, it is instructive to seek a perturbative solution that incorporates the first-order spectral corrections in the nonretarded limit. Expanding Eqs. (3) and (6) around $\omega^{\text{cl}}$, one finds (for a low-loss jellium in free-space)

$$\omega \simeq \omega^{\text{cl}} \left[ 1 - \frac{1}{2} \left( \Upsilon_\perp d_\perp^{(0)} + \Upsilon_\parallel d_\parallel^{(0)} \right) \right], \tag{7a}$$

where $\Upsilon_{\perp,\parallel}$ are geometry- and mode-dependent parameters[17].

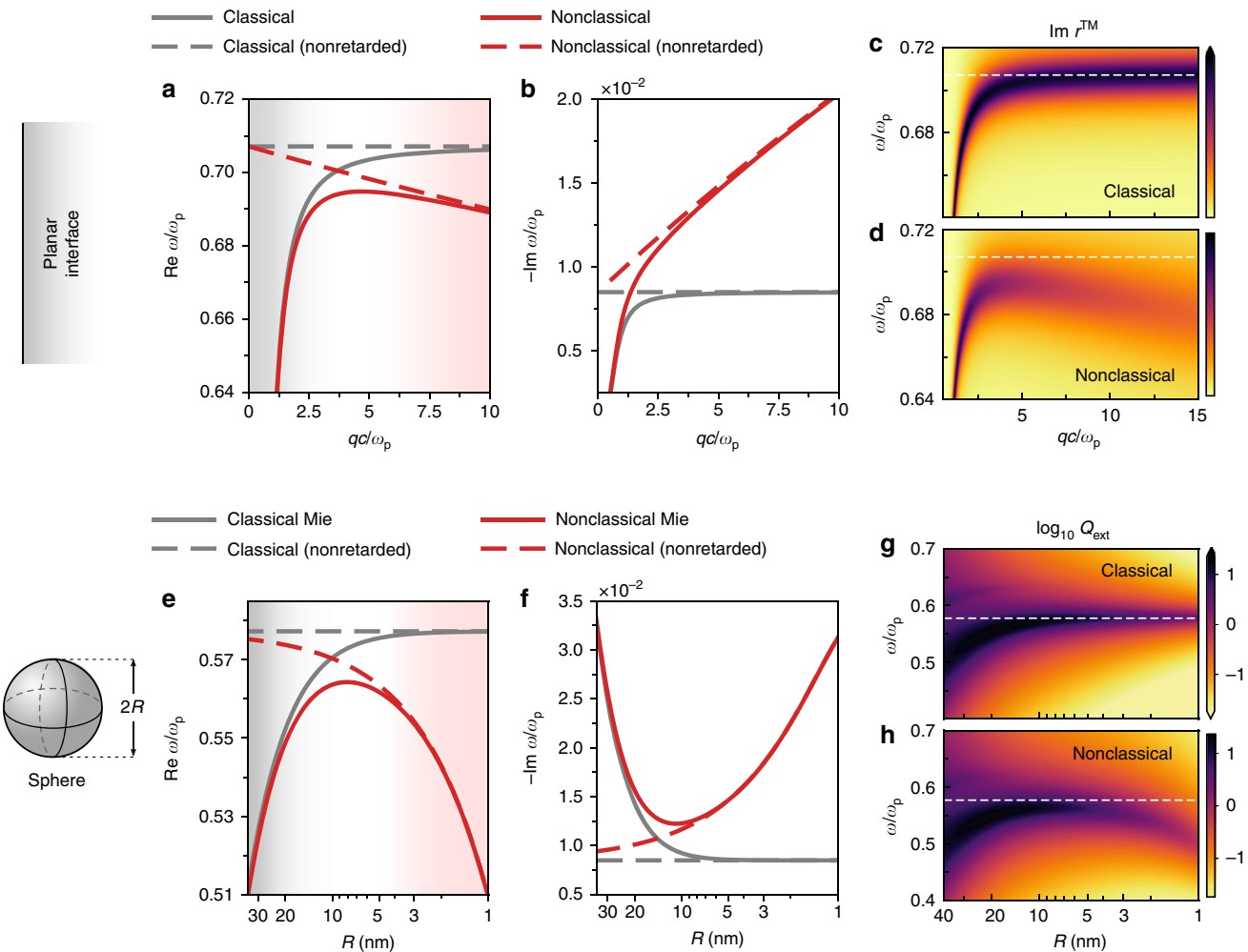

**Fig. 2 Nonclassical corrections to plasmonic spectral properties. a** Dispersion relation of the surface plasmon frequency and **b** resonance width at a planar semi-infinite air–metal interface. **c–d** Im $r^{TM}$ in the classical and nonclassical treatments. Horizontal white dashed lines mark the nonretarded classical surface plasmon at $\omega_p/\sqrt{2}$. **e** Radius-dependent dispersion of the dipole LSP frequency and **f** resonance width in metal spheres. **g** Classical and **h** nonclassical normalized extinction cross-sections [$Q_{ext} \equiv \sigma_{ext}/\pi R^2$] for plasmonic spheres. White dashed lines mark the nonretarded classical dipole resonance at $\omega_p/\sqrt{3}$. Material parameters: jellium metal ($r_s = 4$ and $\hbar\gamma = 0.1\,\mathrm{eV}$) and $\epsilon_d = 1$.

For the planar interface and sphere, they equal

$$\Upsilon_\perp = -\Upsilon_\parallel = \begin{cases} q & \text{planar interface,} \\ (l+1)/R & \text{sphere.} \end{cases} \quad (7b)$$

In the above, $d_{\perp,\parallel}^{(0)} \equiv d_{\perp,\parallel}(\mathrm{Re}\,\omega^{cl})$ is the result of a pole-like approximation. Four points are worth making: (i) the nonclassical correction is directly proportional to an effective $d$-parameter $d_{eff} \equiv d_\perp - d_\parallel$; (ii) the nonclassical frequency shift is approximately proportional to $\mathrm{Re}\,d_{eff}^{(0)}$; (iii) the sign of $\mathrm{Re}\,d_{eff}$ dictates the frequency shift's direction (towards the blue if negative, and towards the red if positive); and (iv) nonclassical broadening due to Landau damping is approximately proportional to $\mathrm{Im}\,d_{eff}^{(0)}$.

The results outlined in this section form the basis for understanding the optical response in the mesoscopic regime, beyond the validity of the classical electrodynamics formulation.

**Nonclassical corrections to the plasmon dispersion.** Figure 2 shows the nonclassical spectral properties of plasmons in a planar (Fig. 2a–d) and spherical (Fig. 2e–h) metallic jellium, contrasting the retarded and nonretarded regimes, as well as the classical and nonclassical behaviors. Figure 2 can thus be regarded as a

corollary of the equations presented in the preceding section. Three (inverse) length scales characterize the plasmonic dispersion in the planar system: the free-space wavevector $k_0$, the plasmon wavevector $q$, and the inverse centroid of induced charge $d_\perp^{-1}$. The plasmon dispersion, consequently, spans up to three distinct regimes, namely a classical, retarded regime $q \sim k_0 \ll |d_\perp|^{-1}$, a deeply nonclassical, nonretarded regime $q \sim |d_\perp|^{-1} \gg k_0$, and an intermediate regime. Figure 2a–d demonstrate that each of these regimes are well-realized in the planar $r_s = 4$ jellium: (i) at small wavevectors, nonclassical effects are negligible; (ii) at large wavevectors, they substantially redshift and broaden the plasmonic dispersion, manifesting the "spill-out" characteristic of simple metals (i.e., $\mathrm{Re}\,d_\perp > 0$) and surface-enhanced Landau damping, respectively, consistent with earlier findings[12,33,39–41]; and (iii) at intermediate wavevectors, both retardation and nonclassical corrections are nonnegligible, and therefore need to be taken into account simultaneously. Intriguingly, the existence of a well-defined intermediate regime demonstrates that the transition from classical to nonclassical response is intrinsically multiscale.

Figure 2e–h outline the plasmonic features of metal spheres as a function of their radii. In most respects, they mirror the qualitative conclusions drawn for the planar case, but with the

inverse radius $R^{-1}$ playing the role of an effective wavevector (increased losses at large radii are due to radiation damping) (see also Supplementary Note 6). Concretely, and focusing on the dipole LSP, Fig. 2e–f show the shortcomings of the classical theory for jellium spheres with dimensions below $2R \sim 20$ nm. For extremely small spheres, the nonretarded limit reproduces the nonclassical redshift and broadening well. Again, we observe an intermediate region where both retardation and nonclassical effects are of comparable magnitude. Notably, this regime has been probed by several experiments that investigated nonclassical plasmons[42–45]. Finally, in Fig. 2g–h we present the normalized extinction cross-sections of jellium spheres under plane-wave illumination. Besides reproducing the main features already observed in Fig. 2e–f, they also exhibit extra resonances due to higher-order LSP modes (Supplementary Fig. S7). The cross-sections associated with these higher-order LSPs, however, fall off rapidly with decreasing radii owing to the realization of the dipole limit. In the nonclassical case this reduction is amplified further, as higher-order LSPs are progressively impacted by surface-induced Landau damping [cf. Eqs. (7a) and (7b)]. These observations are in accord with recent experimental data[42].

The formalism and results presented in the preceding sections establish the fundamentals governing plasmon-enhanced nano-photonic phenomena in the mesoscopic regime. In the following, we exploit this understanding to assess plasmon–emitter interactions at the nanoscale.

**Nonclassical LDOS: Purcell enhancement**. A hallmark of plasmonics is its ability to support extreme field enhancements and correspondingly large Purcell factors[3,19,21], enabling control over the emission properties of emitters. At its core, this is a manifestation of the reshaping of the LDOS spectrum, which is enhanced near plasmon resonances[46–49]. Importantly, the Purcell enhancement is generally maximized at short emitter–surface separations, i.e., exactly where nonlocality and quantum effects become important. Thus, as we show in what follows, a rigorous description of the governing electrodynamics that incorporates nonclassical effects is not only necessary, but essential.

The LDOS, $\rho_{\hat{\mathbf{n}}}^{E}$, experienced by an emitter with orientation $\hat{\mathbf{n}}$ (and incorporating both radiative and nonradiative contributions) is proportional to the imaginary part of the system's Green's dyadic[50], which in turn is expandable in the previously introduced scattering coefficients (see Methods section). We exploit this fact to directly incorporate nonclassical surface corrections into the LDOS, by simply adopting the mesoscopic scattering coefficients, Eqs. (1a) and (1b) or (4a) and (4b), instead of their classical equivalents. In Fig. 3a–b we show the classical and quantum LDOS, normalized to the free-space LDOS, $\rho_0^{E}$, near a planar metal interface as a function of the emitter–metal separation $h$, for a normally-oriented emitter (see Supplementary Note 7 for the parallel and orientation-averaged cases). The enhancement of the LDOS near the surface plasmon frequency is markedly sharper in the classical case and less pronounced in the nonclassical one at shorter separations. This observation is particularly evident in Fig. 3b, which shows the plasmon-enhanced LDOS for different emitter–metal separations. In the classical formulation, the peak in the LDOS remains relatively sharp, approaching the nonretarded plasmon frequency $\omega_p/\sqrt{2}$ at small separations. Contrasting this, in the nonclassical framework the LDOS peak redshifts (consistent with the spill-out characteristic of jellium metals) with decreasing $h$, and evolves into a broad spectral feature at very small emitter–metal distances. This behavior simply reflects the nonclassical corrections to the plasmonic spectrum outlined in the previous section. Evidently, the most significant impact of nonclassicality here is the

substantial reduction (notice the logarithmic scale) of the maximum attainable LDOS in the nonclassical case, particularly for $h \lesssim 10$ nm. Lastly, it is interesting to observe the emergence of a broad spectral peak at frequencies above $\omega_p/\sqrt{2}$ that is absent in the classical setting. This feature is a manifestation of the so-called surface-multipole plasmon or Bennet mode[51] that originates due to the finite-size of the inhomogeneous surface region[33]; mathematically, it corresponds to a pole in $d_\perp(\omega)$; physically, it represents an out-of-plane oscillation confined to the surface region.

Figure 3c–d show the LDOS of a radially-oriented emitter placed at a distance $h$ from the surface of an $R = 5$ nm metal sphere (see Methods section). The LDOS enhancement in the spherical geometry is richer in features, partly because the sphere, unlike the plane, has an intrinsic length scale (its radius $R$), and partly because it hosts a series of $l$-dependent multipolar LSPs. The LDOS enhancement is centered around these LSP frequencies. In the nonclassical case, we again observe redshifted and broadened spectral features relative to their classical counterparts. The impact of Landau damping is amplified by the order of the LSP mode, cf. Eq. (7b); as a result, only the dipole and quadrupole modes are discernible in the nonclassical case (in the classical case, a faint $l = 3$ LSP remains identifiable). Next, in Fig. 3e we investigate the LDOS enhancement's dependence on the sphere's radius $R$ for a fixed emitter–sphere separation of $h = 10$ nm. In particular, the impact of nonclassical effects—particularly its reduction of the maximum LDOS—is more pronounced at smaller radii, in agreement with the approximate $(l + 1)R^{-1}$ scaling previously derived in Eq. (7b). In fact, for very small metal spheres, only the LDOS enhancement associated with the dipole plasmon remains identifiable in the nonclassical case, due to surface-enabled Landau damping. Crucially, although deviations from classicality are most pronounced for spheres with radii $\lesssim 1$ nm, even relatively large spheres (that are otherwise usually considered within the classical regime, e.g., $2R = 50$ nm) exhibit significant nonclassical corrections at small emitter–metal separations. Indeed, this constitutes an example of a multiscale regime where both retardation (a classical effect) and quantum effects must be addressed simultaneously.

**Enhancement of dipole-forbidden multipolar transitions**. The set of optical transitions associated with the emission of radiation by atoms is in practice limited due to the mismatch between the atom's size and the wavelength of the radiation emitted by it. This fact leads to the selection rules for dipole-allowed transitions that originate from the so-called dipole approximation[52]. Such transitions, however, constitute only a fraction of a much richer spectrum. Nevertheless, transition rates other than the dipole-allowed are simply too slow (by several orders of magnitude) to be accessible in practice and are consequently termed "forbidden". Previous works[24,53] have shown that it is possible to increase the effective light–matter coupling strength for such transitions by exploiting, for instance, the shrinkage of the wavelength of light brought about by surface plasmons. Notwithstanding this, a satisfactory framework for describing the impact of nonclassical effects in the plasmonic enhancement of forbidden transitions remains elusive. Below, we remedy this by extending our formalism to the class of dipole-forbidden transitions of electric multipolar character, which can be exploited to probe even larger plasmon momenta. These are transitions in which the orbital angular momentum of the emitter changes by more than one; hereafter denoted E$n$ with $n = 2, 3, 4, \ldots$ (thus, E1 denotes a dipole transition, E2 a quadrupole transition, etc). It should be emphasized that while we consider hydrogenic systems for definiteness in the following, the theory presented here can be

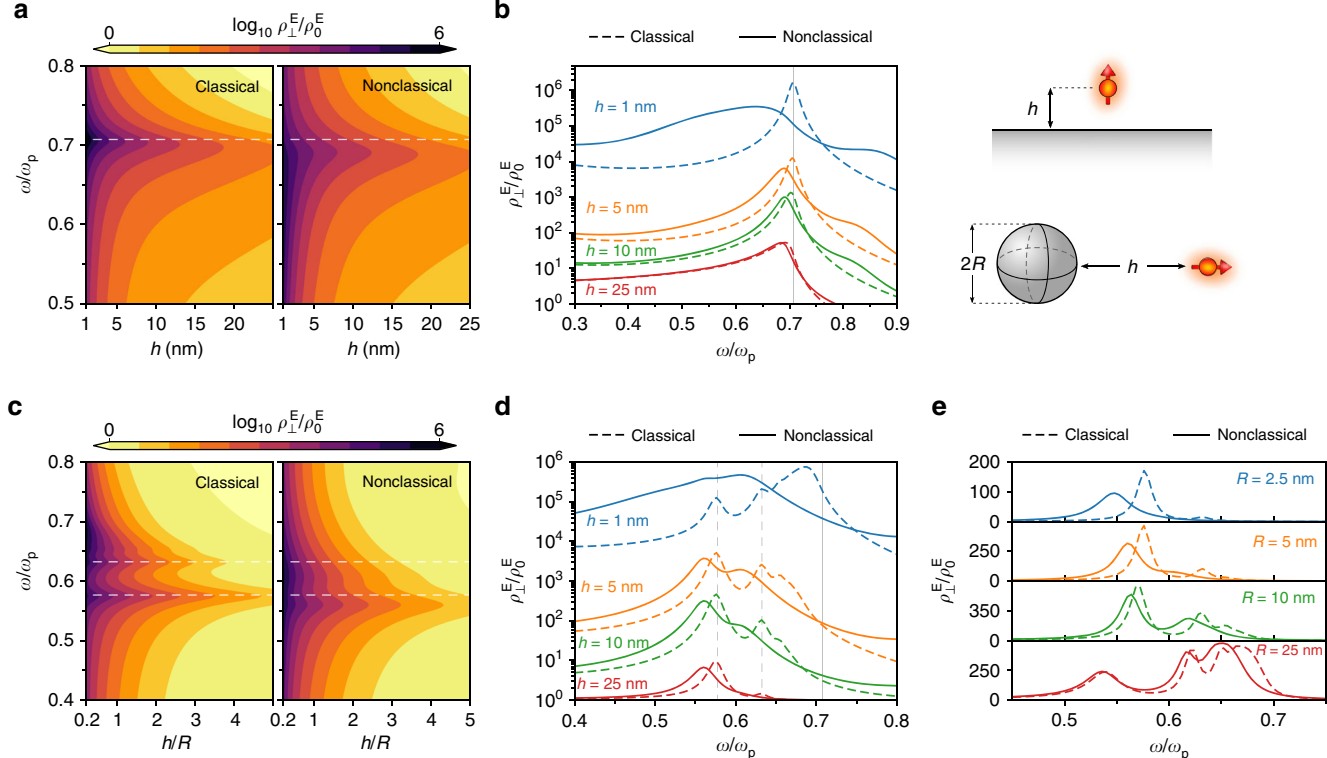

**Fig. 3 Plasmon-mediated Purcell enhancement at the nanoscale.** Normalized LDOS, $\rho^{E}/\rho_0^{E}$, of a normally-oriented emitter near **a–b** a planar metal surface and **c–e** a metal sphere. **a** Normalized LDOS as a function of the emitter's frequency and separation from the jellium surface. The horizontal line marks $\omega_p/\sqrt{2}$. **b** LDOS enhancement at different emitter–surface separations. The vertical line marks $\omega_p/\sqrt{2}$. **c** Normalized LDOS near an $R = 5$ nm jellium sphere as a function of the emitter's frequency and $h/R$ ratio. Horizontal lines mark the nonretarded classical LSP frequencies $\omega_p/\sqrt{1+(l+1)/l}$ for $l \in \{1, 2\}$. **d** LDOS enhancement near an $R = 5$ nm sphere for different emitter–surface distances. Vertical lines mark the $l = 1$, $l = 2$, and $l = \infty$ nonretarded classical LSPs. **e** Normalized LDOS for plasmonic spheres of differing radii at a fixed $h = 10$ nm emitter–surface separation. Material parameters as in Fig. 2.

readily applied to any point-like emitter (e.g., atoms, quantum dots, nitrogen-vacancy centers, or dyes).

We consider an emitter at a distance $h$ from a planar metal surface (Fig. 4a), and treat the light–matter interaction in its vicinity using a formulation of macroscopic quantum electrodynamics which enables a rigorous account of the quantum nature of the emitter and of the plasmon, and the inherent presence of loss[54,55]. Within this framework, the multipolar decay rates, $\Gamma_{En}$, can be evaluated as[24] (Supplementary Note 8)

$$\Gamma_{En} = 2\alpha^3 \omega_0 \left[ \frac{(k_0 a_B)^{n-1}}{(n-1)!} \right]^2 \Xi \int_0^\infty u^{2n} e^{-2uk_0 h} \operatorname{Im} r^{\mathrm{TM}} \, \mathrm{d}u, \quad (8)$$

where $u \equiv q/k_0$, $a_B$ denotes the Bohr radius, $\alpha$ is the fine-structure constant, and the dimensionless quantity $\Xi$ is related to the matrix element associated with the transition (Supplementary Note 8). In the previous expression, the nonretarded limit is assumed, valid for $k_0 h \ll 1$. Nonetheless, in our calculations we use the retarded reflection coefficient to accurately incorporate the plasmon pole's spectral position. Moreover, in this limit $\Gamma_{En}^{\mathrm{tot}} = \Gamma_{En}^0 + \Gamma_{En} \simeq \Gamma_{En}$ since the free-space contribution $\Gamma_{En}^0$ is many orders of magnitude smaller.

In Fig. 4b we plot the E$n$ decay rates associated with the 6{p, d, f, g, h} → 2s transition series in hydrogen ($\delta$-transitions of the Balmer series). While at relatively large distances from the metal the spontaneous emission rates of higher-order multipolar transitions are several orders of magnitude slower than E1, this difference is dramatically reduced at smaller emitter–metal separations. Interestingly, at nanometric separations the higher-order multipolar rates can exceed the E1 free-space rate, signaling

a breakdown of traditional dipole-allowed selection rules. More interesting still, the inclusion of nonclassical effects via $d$-parameters increases the multipolar decay rates relative to the classical predictions (Fig. 4b, inset), by roughly one order of magnitude at the smallest separations. To understand the physical mechanism for this additional enhancement, we show in Fig. 4c–e the integrand of Eq. (8) for the first three multipolar orders, each evaluated at three distinct atom-metal separations. Two main contributions can be readily identified: (i) a sharp, resonant contribution corresponding to the plasmon pole embodied in $\operatorname{Im} r^{\mathrm{TM}}$ at the transition frequency (i.e., at the intersection of the blue and red lines in Fig. 4a), associated with emission into plasmons; and (ii) a broad, nonresonant contribution associated with quenching by lossy channels in the metal, e.g., Landau damping, disorder, phonons, etc. The relative contribution of (i) and (ii) to the overall decay rate depends strongly on the emitter–metal separation (due to the $u^{2n} e^{-2uk_0 h}$ scaling of the integrand), with loss-related quenching dominating over plasmon emission at very small emitter–metal separations. This effect is more pronounced for higher-order multipolar transitions since the integrand of Eq. (8) initially grows with $u^{2n}$. The additional nonclassical enhancement is thus readily understood: it is a direct result of an increased nonresonant, loss-related contribution due to surface-enabled Landau damping. Finally, the dotted lines in Fig. 4b, f indicate regions in which a significant fraction of $\Gamma_{En}$ is accumulated at very large wavevectors where the condition $q \operatorname{Re} d_\perp \ll 1$ is only approximately valid; evidently, at the smallest separations and at large transitions orders $n$, our mesoscopic framework is pushed beyond its range of validity.

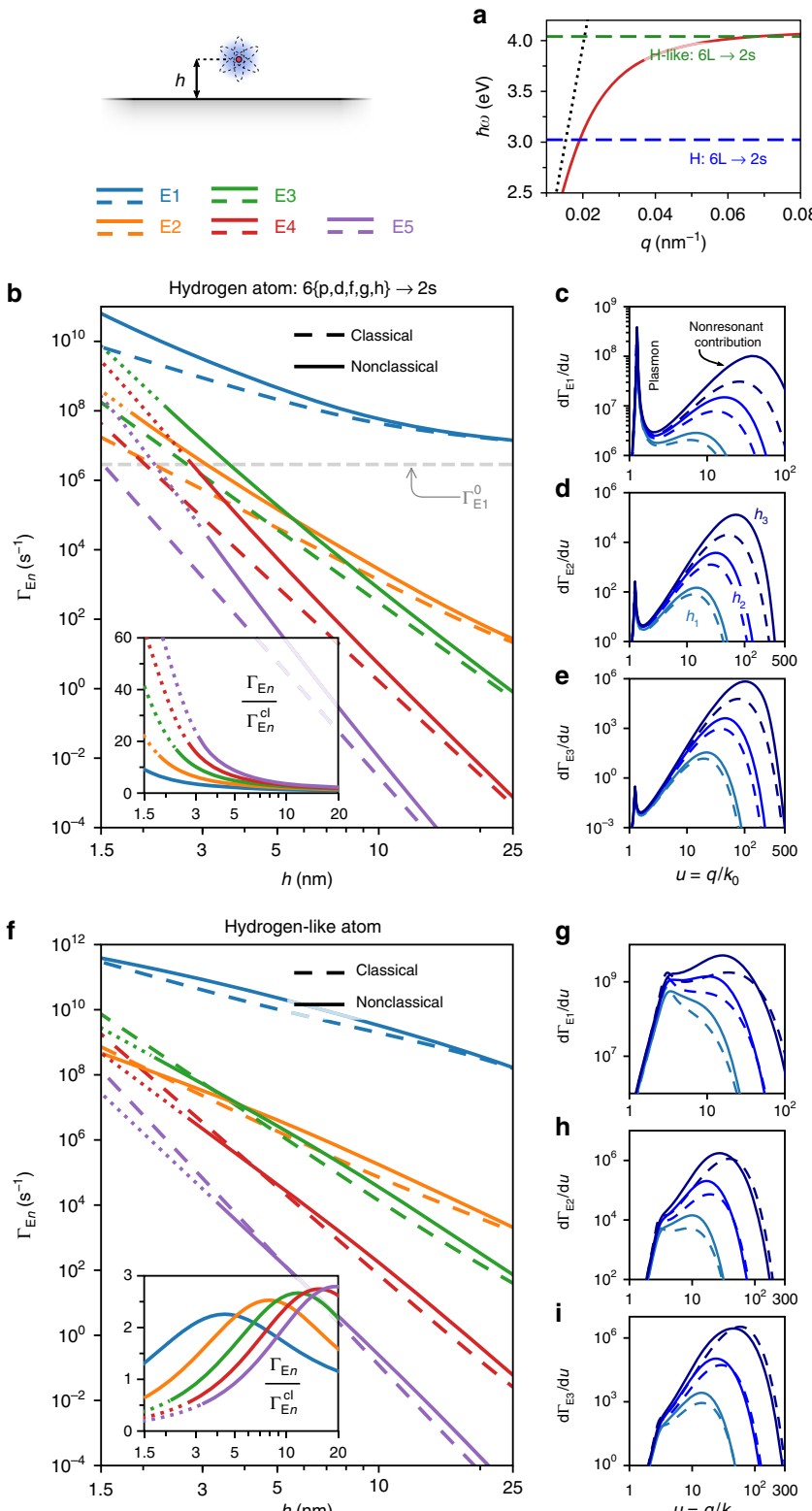

**Fig. 4 Enhancement of dipole-forbidden electric multipole transitions. a** Transition frequencies considered in **b–e** (dashed blue) and **f–i** (dashed green) relative to the plasmon (red) and light-line (dotted black) dispersion. **b** Multipolar transition rates, $\Gamma_{En}$, associated with the 6{p, d, f, g, h} → 2s series of transitions in hydrogen ($\hbar\omega_0 = 3.02$ eV). The dashed gray horizontal line marks the free-space E1 rate. The inset shows the relative rates, $\Gamma_{En}/\Gamma_{En}^{cl}$. **c–e** Differential rates $d\Gamma_{En}/du$ [integrand of Eq. (8)] for various atom–surface separations: $h_1 = 10$ nm, $h_2 = 5$ nm, and $h_3 = 2.5$ nm. **f–i** Equivalents of **b–e** but at the transition frequency $\omega_0 = 0.97\,\omega_p/\sqrt{2}$. Dotted lines indicate regions where our nonclassical framework is pushed beyond its range of validity (identified heuristically as cases where contributions to $\Gamma_{En}$ from $q_{th}\mathrm{Re}\,d_\perp \geq 1/3$ exceeds 10%).

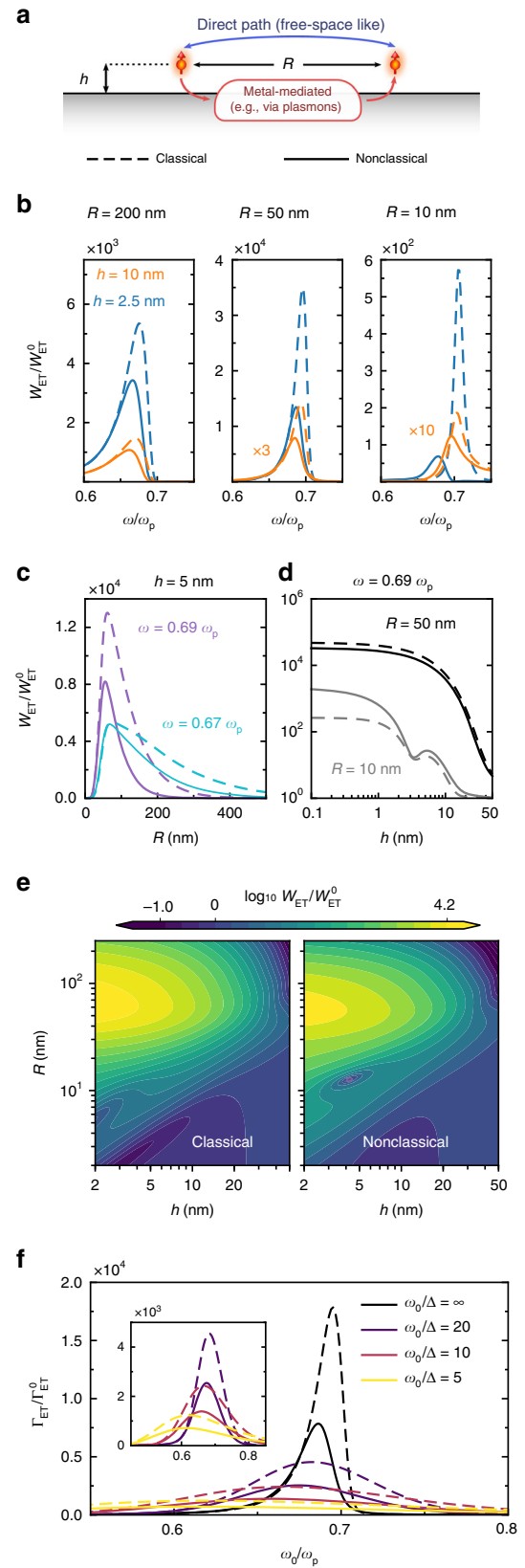

**Fig. 5 Nonclassical corrections to plasmon-mediated energy transfer near a planar metal. a** Two emitters transfer energy through free-space and metal-mediated channels (mutual separation $R \equiv |\mathbf{r}_A - \mathbf{r}_D|$, metal-interface offset $h = z_D = z_A$, and normally orientated, i.e., $\boldsymbol{\mu}_A \parallel \boldsymbol{\mu}_D \parallel \hat{\mathbf{z}}$). **b** Spectral dependence of the (normalized) ET amplitude, $w_{ET}/w_{ET}^0$, for varying emitter–emitter distances. **c, d** Normalized transfer amplitude as a function of $R$ ($h$), for fixed $h$ ($R$) (see labels). **e** Contours of the classical and nonclassical energy transfer amplitudes at $\omega_0 = 0.97\,\omega_p/\sqrt{2}$. **f** Plasmonic enhancement of ET rates ($h = 5$ nm and $R = 50$ nm) for broadband emitters of varying $Q = \omega_0/\Delta$ as a function of (joint) emitter frequency $\omega_0$ (inset: finite-$Q$ emitters only).

series. The enhancement of the E$n$ rates is qualitatively similar to the previous case, albeit with some quantitative differences: for instance, as shown in Fig. 4g–i, the resonant plasmon contribution now peaks at larger $u$; a simple consequence of the increased plasmon momentum at this higher transition frequency. This is in principle beneficial because even a small increase in confinement can result in a huge increase of the decay rates due to the $u^{2n}$ scaling of $d\Gamma_{En}/du$. However, plasmon losses tend to increase concomitantly with increasing confinement, resulting in broader plasmon peaks (cf. Fig. 4g–i). Lastly, we observe that the nonclassical multipolar decay rates no longer consistently exceed the classical predictions at this higher frequency, contrasting our findings in Fig. 4b. This difference reflects a more complicated and substantial nonclassical modification of the plasmonic response at such frequency (see Fig. 2d). The overall impact on $\Gamma_{En}$ ultimately results from an nontrivial interplay between the modified scattering response $\mathrm{Im}\,r^{TM}$ and the scaling $u^{2n}e^{-2uk_0h}$.

Our calculations demonstrate that quantum surface corrections substantially modify the multipolar decay rates from those predicted in classical electrodynamics; especially off-resonance, where the discrepancy increases with the multipolar transition order. Radiation from these multipolar transitions can be delivered to the far-field by outcoupling the SPPs via gratings or antennas. Moreover, even in the regime dominated by nonresonant enhancement, the breakage of the conventional selection rules should still have clear experimental signatures, with potential implications for photovoltaic devices[56] or hot-electron catalysis[56,57].

**Energy transfer between two emitters**. The interaction between emitters in optical cavities or near plasmonic structures is instrumental to many scientific disciplines, ranging from quantum optics[58] to chemical physics and the life sciences[59,60]. A prominent example is energy transfer (ET) between two fluorophores: the fundamental process by which an excited flourophore (the donor, D) lowers its energy by transferring it to another flourophore (the acceptor, A). The signature of this mechanism is the observation of fluorescence emitted by the acceptor. In free-space, the ET between the two emitters takes place primarily via dipole–dipole interaction and is typically short-ranged; in this limit, it is commonly referred to as Förster resonant energy transfer (FRET). Here too, the integration of emitters with plasmonic nanostructures can enhance the emitter–emitter ET rate, $\Gamma_{ET}$, through the introduction of a new, plasmonic near-field channel between the donor (D) and the acceptor (A)[61–63].

With this in mind, we investigate the impact of nonclassical corrections to plasmon-mediated ET between two emitters near a planar metal surface (Fig. 5a). The calculation of $\Gamma_{ET}$ involves the system's Green's function **G** (Supplementary Note 9), which in turn depends on the system's scattering coefficients. Concretely,

Figure 4f considers a similar transition in a hydrogen-like atom, but now occurring at a higher frequency—i.e., closer to $\omega_p/\sqrt{2}$—and thus probing larger plasmon wavevectors. We assume, for simplicity, that the magnitude of the matrix elements in Eq. (8) still equal those in the 6{p, d, f, g, h} → 2s hydrogen

for two emitters above a metal surface, the ET rate from a donor located at $\mathbf{r}_D$ to an acceptor placed at $\mathbf{r}_A$ can be determined via[25–27,47,50]

$$\Gamma_{ET} = \int_0^\infty w_{ET}(\mathbf{r}_D, \mathbf{r}_A; \omega) f_D^{em}(\omega) f_A^{abs}(\omega) \, d\omega, \qquad (9)$$

where $w_{ET}(\mathbf{r}_D, \mathbf{r}_A; \omega) \equiv \frac{2\pi}{\hbar^2} \left(\frac{\omega^2}{\epsilon_0 c^2}\right)^2 \left|\boldsymbol{\mu}_A^* \cdot \mathbf{G}(\mathbf{r}_D, \mathbf{r}_A; \omega) \cdot \boldsymbol{\mu}_D\right|^2$ is the ET amplitude, which governs the medium-assisted interaction. Here, $f_D^{em}$ ($f_A^{abs}$) is the donor's emission (acceptor's absorption) spectrum, and $\boldsymbol{\mu}_D$ ($\boldsymbol{\mu}_A$) the corresponding dipole moment.

Figure 5b–e show the ET amplitude $w_{ET}(R, \omega)$ (evaluated at $z_A = z_D \equiv h$ with a donor–acceptor separation $|\mathbf{r}_A - \mathbf{r}_D| \equiv R$) normalized to its value in free-space $w_{ET}^0(R, \omega)$. The advantage of such procedure is that this ratio is emitter-independent, facilitating a discussion on the impact of the plasmonic response (also, for spectrally aligned narrowband emitters where $f_D^{em}(\omega) f_A^{abs}(\omega) \sim \delta(\omega - \omega_0)$, this simply amounts to the total ET rate enhancement $\Gamma_{ET}/\Gamma_{ET}^0$; we shall return to this point below). Our results demonstrate that the omission of quantum mechanical effects leads to a significant overestimation of the normalized ET amplitudes, across a broad parameter space. This discrepancy is particularly pronounced for emitter–metal separations of about $h \lesssim 10$–15 nm, and spans a wide range of donor–acceptor separations, $R$. The ET dependence on $R$ is particularly interesting and spans several distinct regimes: (i) for large $R$ relative to the SPP's propagation length, $L_p$, the metal's impact is negligible [the emitters are simply too far away for the ET to be mediated by surface plasmons (i.e., a SPP excited by the donor will be dissipated long before it reaches the acceptor)]; (ii) for $R \sim L_p$, the ET enhancement reaches its maximum, whose position and value are dictated by the spectral properties of the SPP, and therefore is affected both by the nonclassical spectral shift and broadening; and (iii) for $R \ll h$, the interaction is dominated by the free-space channel, rendering the metal's impact negligible again.

For emitters of sufficient spectral width, ET can assume a broadband aspect: we explore this in Fig. 5f by computing $\Gamma_{ET}/\Gamma_{ET}^0$ for a Gaussian donor–acceptor overlap $f_D^{em}(\omega) f_A^{abs}(\omega) = e^{-(\omega-\omega_0)^2/2\Delta^2}/\sqrt{2\pi}\Delta$, centered at $\omega_0$ and with a (joint) width $\Delta$ and quality factor $Q \equiv \omega_0/\Delta$. Figure 5f shows the normalized classical and nonclassical broadband integrated ET rates for several $Q$ as a function of the center frequency $\omega_0$. Clearly, the maximum of $\Gamma_{ET}/\Gamma_{ET}^0$ decreases with $Q$, with a concomitant broadening and redshifting of the central peak. Interestingly, though the highest ET rate enhancements are obtained at large $Q$, and for $\omega_0$ near the SPP's resonance, this shows that spectrally misaligned emitters can benefit from small $Q$ factors, as this extends their spectral tails into the resonant region. More importantly, our results show that nonclassicality remains important even in the case of broadband emitters, and that nonclassical deviations persist (after being broadband integrated) even when the joint spectral width is larger than the nonclassical plasmon resonance shift.

Lastly, Fig. 5 demonstrates the importance of accounting for nonclassical effects in ET, which impose limits to the maximum attainable plasmon-enhanced ET rate between emitters.

**Plasmon-enhanced two-photon emission.** The emission of light by an excited emitter is generally very well-described by first-order perturbation theory in the light–matter interaction described by quantum electrodynamics (including every process considered so far), reflecting its intrinsic weakness. At higher order in the interaction, the possibility of two- and multi-photon spontaneous emission emerges. While two-photon spontaneous emission was predicted as early as 1931 by Göppert-Mayer[64], it eluded observation for decades in both atomic and solid-state systems[28,29], due to the exacerbated weakness of the interaction at second order. Despite this, two-photon emission is an attractive process due to the correlated nature of the emitted photons (entangled in e.g., energy and angular momentum). The extreme nanoscale confinement of plasmons in metals provides new opportunities to enhance two-photon emission dramatically[30] (in the guise of two-plasmon emission), with recent work identifying opportunities to enhance two-photon emission to be as strong[24], or even far stronger[65], than single-photon emission. However, with these possibilities being enabled essentially by extreme nanoscale confinement, it is natural to anticipate a sizable impact of nonclassical effects.

A minimal model of two-photon spontaneous emission is shown in Fig. 6a, where we illustrate an emitter at a distance $h$ from a semi-infinite metallic interface. To isolate the parts of two-photon emission that depend on the metallic interface, as opposed to the detailed atomic level structure, we consider two-photon transitions between the s-states of a simple hydrogenic atom. This subgrouping includes the most prominent example of two-photon emission: the 2s → 1s transition in hydrogen, with level separation $\omega_0 = \omega_{2s} - \omega_{1s} \approx 10.2$ eV. The level separation $\omega_0$ restricts the frequencies of the two emitted photons to $\omega \in ]0, \omega_0[$ and $\omega' \equiv \omega_0 - \omega$ (reflecting energy conservation) but otherwise leaves their difference unconstrained. The emission process is consequently broadband, with the total rate $\Gamma_{TPE}$ a summation of all energy-allowed $(\omega, \omega')$-pairs: $\Gamma_{TPE} = \int_0^{\omega_0} (d\Gamma_{TPE}/d\omega) \, d\omega$, where $d\Gamma_{TPE}/d\omega$ is the differential decay rate for two-photon emission into frequencies $\omega$ and $\omega_0 - \omega$. As an example, for the 2s → 1s transition of hydrogen in free-space, $d\Gamma_{TPE}^0/d\omega$ exhibits a broad peak around the equal $\omega = \omega' = \omega_0/2$ splitting, as shown in Fig. 6b. Its integral, corresponding to the decay rate, is about 8.3 s$^{-1}$, nearly eight orders of magnitude slower than the 2p → 1s dipole-allowed single-photon transition ($\approx 6.3 \times 10^8$ s$^{-1}$)[66].

In the presence of a metallic interface, the situation changes drastically, due to a strongly enhanced LDOS. In fact, two-photon emission benefits twice from an enhanced LDOS, encoded by the following nonretarded expression[65] for the enhancement of the differential decay rate $d\Gamma_{TPE}/d\omega$ for an s → s transition in a hydrogenic atom, relative to its free-space value $d\Gamma_{TPE}^0/d\omega$ (Supplementary Note 9):

$$\frac{d\Gamma_{TPE}/d\omega}{d\Gamma_{TPE}^0/d\omega} = \frac{1}{2}\left(\frac{\rho_\perp^E(\omega)}{\rho_0^E(\omega)}\right)\left(\frac{\rho_\perp^E(\omega_0 - \omega)}{\rho_0^E(\omega_0 - \omega)}\right). \qquad (10)$$

Each fraction is a Purcell factor; thus, the order of magnitude two-photon differential enhancement is roughly the square of the one-photon enhancement (Fig. 3). More precisely, the differential two-photon enhancement is directly and simply related to the one-photon enhancement: it is (half) the Purcell enhancement at $\omega$ multiplied by its reflection about $\omega_0/2$.

We note that the computation of $\Gamma_{TPE} = \int_0^{\omega_0}(d\Gamma_{TPE}/d\omega) \, d\omega$ for $\omega_0 > \omega_p$ in a nonclassical setting requires knowledge of $d_\perp(\omega)$ above the plasma frequency (similarly so for ET when $f_D^{em}(\omega) f_A^{abs}(\omega)$ extends above $\omega_p$, see Eq. (9)). Direct calculation of $d_\perp(\omega)$ via TDDFT is cumbersome above $\omega_p$, since the induced potential extends into the bulk; instead, following refs. [67,68], we extrapolate $d_\perp(\omega)$ to $\omega > \omega_p$ by enforcing exact sum rules and asymptotic limits on a fit of $d_\perp(\omega)$ over frequencies below $\omega_p$ (Supplementary Note S9).

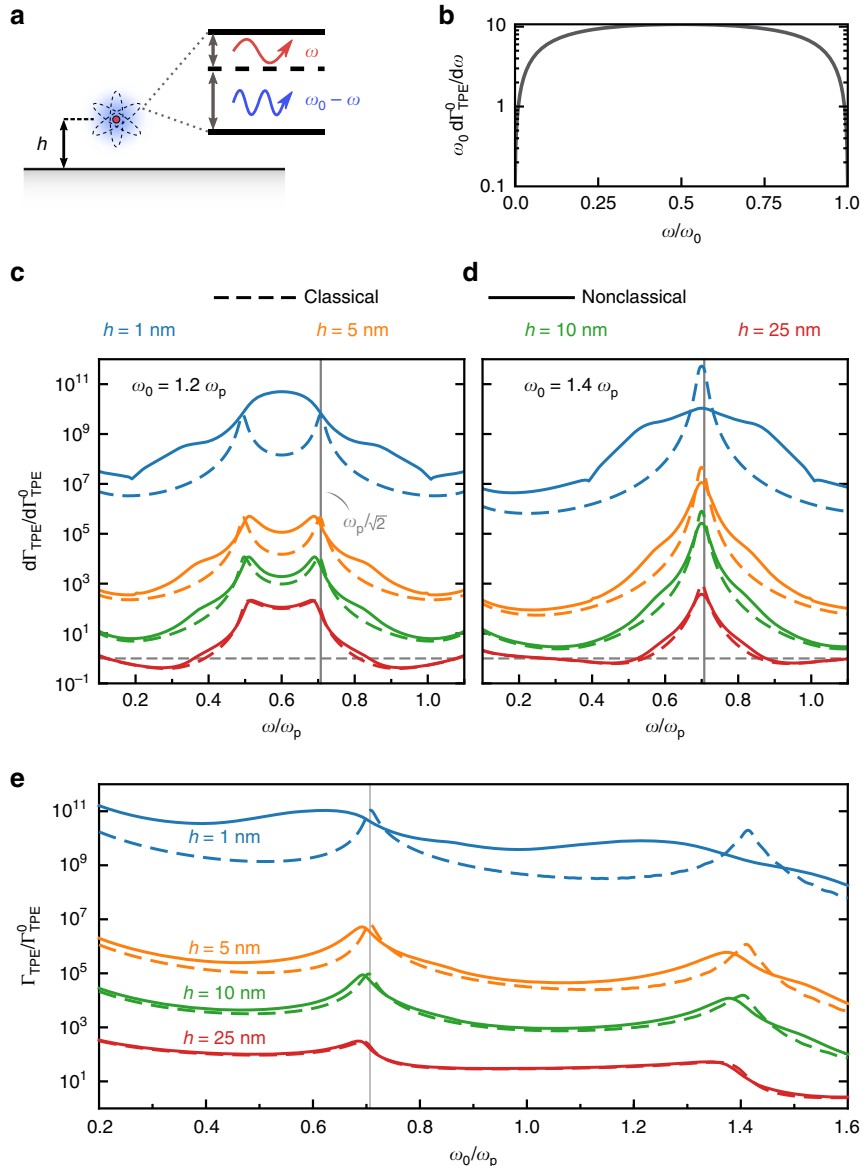

**Fig. 6 Nonclassical corrections to two-photon emission enhancement. a** Two-photon emission (s → s transition) of a hydrogenic atom above a planar air–metal interface. **b** Differential decay rate in free-space for the 2s → 1s two-photon transition in hydrogen. **c**, **d** Enhancement of the differential decay rate $d\Gamma_{TPE}/d\omega$, Eq. (10), as a function of frequency for different emitter–interface separations at a transition frequency of $\omega_0 = 1.2\omega_p$ ($1.4\omega_p$). **e** Enhancement of the integrated two-photon decay rate $\Gamma_{TPE}$, as a function of transition frequency and emitter–surface separation.

Figure 6c–d contrast the classical and nonclassical predictions of the differential two-photon emission enhancement near a metal surface for different values of the (hydrogen-like emitter's) transition frequency, its separation from the surface, and emission frequency. For separations ≳ 10 nm nonclassical effects modify the physics quantitatively, but not qualitatively. Deviations from classicality substantially increase at the separation of 5 nm, with clear hallmarks of nonclassical broadening in particular. At a 1 nm separation, the classical and nonclassical predictions differ qualitatively: at the transition frequency $\omega_0 = 1.2\omega_p$ (Fig. 6c) the peak-structure and position is mostly dissimilar (as can be understood and expected from Fig. 3b, where the LDOS peak is similarly displaced from the classical prediction); at $\omega_0 = 1.4\omega_p$ (Fig. 6d), the classical and nonclassical peak positions still coincide but the nonclassical spectrum is far broader.

Finally, the impact of nonclassicality on the enhancement of the total (i.e., integrated) two-photon decay rate is shown in Fig. 6e. For small separations, the classical prediction can be quantitatively

inaccurate by an order of magnitude. However, as also seen in the case of the LDOS, the classical prediction does not necessarily lead to an overestimation of the decay rates: for some transition frequencies, the nonclassical decay rate is higher, due to a redistribution of LDOS into regions in which the classical LDOS was low. Due to the quadratic dependence of two-photon emission enhancement on the LDOS, this process is much more sensitive to deviations from classicality. The considerations of two-photon emission in this section provides yet another example of the substantial impact of nonclassical effects to nanoscale plasmon–emitter interactions.

## Discussion

In this article, we have considered the impact of nonclassical corrections in a varied range of plasmon-enhanced light–matter interaction processes using a scattering framework that incorporates nonclassical effects via Feibelman $d$-parameters. These plasmon-empowered processes include spontaneous dipole and multipole emission, ET between emitters, and spontaneous

two-photon emission. Our findings elucidate and contextualize the main physical mechanisms responsible for deviations from the classical response in light–matter interactions at the nanoscale: spectral shifting and surface-enabled Landau damping, manifesting the joint impact of spill-out and nonlocality. For deeply nanoscale emitter–surface separations, e.g., below $\sim 5$ nm, the deviations can be order-of-magnitude, thus completely invalidating any quantitative aspect of the classical approach.

There are several interesting opportunities and open questions arising from this work. First, our approach can be readily extended to other prominent light–matter interaction processes, such as near-field radiative heat transfer[69], electron energy loss spectroscopy[42,45], or van der Waals[70] and Casimir–Polder interactions[71]. Second, the $d$-parameter framework is agnostic of the model employed to calculate the $d$-parameters. Here, we have employed jellium TDDFT, but other models, such as hydrodynamic response (within the hydrodynamical model (HDM), the $d$-parameters of a homogeneous electron gas adjacent to vacuum are $d_\perp^{\mathrm{HDM}}(\omega) = -\beta/(\omega_{\mathrm{p}}^2 - \omega^2)^{1/2}$ and $d_\parallel^{\mathrm{HDM}}(\omega) = 0$, with $\beta^2 = 3v_{\mathrm{F}}^2/5$)[16], can be readily treated by $d$-parameters as well. Similarly, the jellium approximation can be relaxed in atomic TDDFT, posing new, fundamental questions—particularly pertinent in noble metals—on the role of atomic structure and valence-band bound screening. Third, recent experiments have demonstrated that the $d$-parameters can be directly inferred from far-field optical measurements[36]: comparison between measurements of plasmon-enhanced light–matter interaction at the nanoscale and theoretical predictions, such as those detailed here, could open a complementary avenue for experimental characterization of $d$-parameters. We emphasize that the nanometer-scale emitter–surface separations that lead to substantial quantum corrections in light–matter interactions are well-within the reach of current experimental capabilities[3,19,36,72]. Indeed, several earlier experiments[18,73] have probed the requisite parameter regimes; their observations of deviations from classical predictions may already suggest evidence of the corrections described here. Fourth, the formalism presented here can be readily applied in arbitrary geometries via $d$-parameter-modified mesoscopic boundary conditions (Supplementary Note 2)[36]. Fifth and lastly, while we have restricted our focus to quantum corrections due to the plasmonic surface-response, a separate class of corrections exist with origin in the emitter, emerging beyond the point-emitter approximation[74–77]. Our framework can be readily adapted to include these emitter-centric corrections (Supplementary Note 8); that prospect is particularly interesting in "large" emitters, such as quantum dots or molecules, where their magnitude can be substantial.

Realizing the promise of plasmon-enhanced light–matter interactions inevitably involves multiscale plasmonic architectures, combining both wavelength- and nanoscale features in synergy. The development of the next generation of nanoscale optical devices consequently requires new theoretical tools that incorporate the salient features of both the classical and quantum domains in a tractable manner: the mesoscopic framework developed here constitutes such a tool.

## Methods

**Feibelman $d$-parameters**. The complex-valued Feibelman $d$-parameters, $d_\perp$ and $d_\parallel$, can be defined from the quantum mechanical induced charge density, $\rho(\mathbf{r}) \equiv \rho(z)e^{iqx}$, and associated induced current density, $\mathbf{J}(\mathbf{r}) \equiv \mathbf{J}(z)e^{iqx}$ (frequency-dependence suppressed, but implicit)[16,17,33]:

$$d_\perp = \frac{\int_{-\infty}^{\infty} z\rho(z)\,\mathrm{d}z}{\int_{-\infty}^{\infty} \rho(z)\,\mathrm{d}z} \quad \text{and} \quad d_\parallel = \frac{\int_{-\infty}^{\infty} z\partial_z J_x(z)\,\mathrm{d}z}{\int_{-\infty}^{\infty} \partial_z J_x(z)\,\mathrm{d}z}, \quad (11)$$

here, for an interface spanning the $xy$-plane at $z = 0$. It is apparent from Eq. (11) that $d_\perp$ corresponds to the centroid of the induced charge density (cf. Fig. 1), while

$d_\parallel$ corresponds to the centroid of the normal derivative of the tangential current (which is identically zero for charge-neutral interfaces)[33]. Unlike the bulk permittivity that characterizes a single material, the $d$-parameters are surface response functions that depend on the two materials that make up the interface (including, in principle, their surface terminations). Here, we restricted our focus to the vacuum–jellium interface.

In short, the essential appeal of $d$-parameters is their facilitation of a practical introduction of the important electronic length scales associated with the dynamics of the electron gas at an interface (Supplementary Note 1).

**LDOS calculations**. The LDOS experienced by a point-like dipole emitter embedded in a dielectric medium with dielectric constant $\epsilon_{\mathrm{d}}$ and located at a distance $h$ above a metal half-space is given by (see also Supplementary Note 7)[50]

$$\frac{\rho_\perp^{\mathrm{E}}}{\rho_0^{\mathrm{E}}} = 1 + \frac{3}{2}\,\mathrm{Re}\int_0^\infty \frac{u^3}{\sqrt{1-u^2}}\, r^{\mathrm{TM}} e^{2ik_{\mathrm{d}}h\sqrt{1-u^2}}\,\mathrm{d}u, \quad (12a)$$

$$\frac{\rho_\parallel^{\mathrm{E}}}{\rho_0^{\mathrm{E}}} = 1 + \frac{3}{4}\,\mathrm{Re}\int_0^\infty \frac{u}{\sqrt{1-u^2}}\big[r^{\mathrm{TE}} - (1-u^2)r^{\mathrm{TM}}\big] e^{2ik_{\mathrm{d}}h\sqrt{1-u^2}}\,\mathrm{d}u, \quad (12b)$$

for an emitter with its dipole moment oriented perpendicular ($\perp$) or parallel ($\parallel$), respectively, to the dielectric–metal interface (here, the $z = 0$ plane). The perpendicularly oriented dipole only couples to TM modes, whereas the dipole in the parallel configuration couples to both TM and TE modes. At short emitter–metal separations, however, the TM contribution dominates, regardless of orientation. Moreover, since plasmonic excitations are TM polarized, the TM contribution is the main quantity of interest for plasmon-enhanced LDOS.

For an emitter at a distance $h$ from the surface of a metallic sphere of radius $R$, the LDOS can be evaluated via (Supplementary Note 7)[37,78]

$$\frac{\rho_\perp^{\mathrm{E}}}{\rho_0^{\mathrm{E}}} = 1 + \frac{3}{2}\frac{1}{y^2}\sum_{l=1}^\infty (2l+1)l(l+1)\,\mathrm{Re}\Big\{-a_l^{\mathrm{TM}}[h_l^{(1)}(y)]^2\Big\}, \quad (13a)$$

$$\frac{\rho_\parallel^{\mathrm{E}}}{\rho_0^{\mathrm{E}}} = 1 + \frac{3}{4}\frac{1}{y^2}\sum_{l=1}^\infty (2l+1)\,\mathrm{Re}\Big\{-a_l^{\mathrm{TM}}[\xi_l'(y)]^2 - a_l^{\mathrm{TE}}[\xi_l(y)]^2\Big\}, \quad (13b)$$

for an emitter with its dipole oriented along the radial ($\perp$) or tangential ($\parallel$) directions, respectively. In addition, we have introduced the dimensionless radial emitter position $y \equiv k_{\mathrm{d}}(R + h)$ for brevity of notation.

The above expressions also highlight a key feature exploited in all our calculations: conveniently, in order to calculate the quantum mechanically corrected LDOS within the $d$-parameters framework one only needs to replace the standard Mie coefficients by their generalized nonclassical counterparts, Eqs. (4a) and (4b). The same also holds for the standard Fresnel reflection coefficients and their nonclassical counterparts, Eqs. (1a) and (1b).

## Data availability

The data that underlie the findings of this study are available from the corresponding authors upon reasonable request.

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

## Acknowledgements

P.A.D.G. acknowledges fruitful discussions with Martijn Wubs on Förster-type energy transfer. The Center for Nanostructured Graphene is sponsored by the Danish National Research Foundation (Project No. DNRF103). T.C. acknowledges support from the Danish Council for Independent Research (Grant No. DFF–6108-00667). N.R. recognizes the support of the DOE Computational Science Graduate Fellowship (CSGF) fellowship No. DE-FG02-97ER25308. N.A.M. is a VILLUM Investigator supported by VILLUM FONDEN (Grant No. 16498) and Independent Research Fund Denmark (Grant No. 7079-00043B). Center for Nano Optics is financially supported by the University of Southern Denmark (SDU 2020 funding). This work was partly supported by the Army Research Office through the Institute for Soldier Nanotechnologies under contract No. W911NF-18-2-0048, as well as in part by the MRSEC Program of the National Science Foundation under Grant No. DMR-1419807.

## Author contributions

All authors contributed to all aspects of this work.

## Competing interests

The authors declare no competing interests.
