## [Peer Review File · Nature Communications]

Reviewers' Comments:

Reviewer #3:

Remarks to the Author:

In this paper, the authors use a theoretical treatment of mesoscopic electrodynamics based on Feibelman d-parameters to investigate the effect of nonclassical effects on a broad array of plasmon-emitter interactions, including dipolar and multipolar spontaneous emission enhancement, plasmon-assisted energy transfer, and enhancement of two-photon transitions. This topic is of broad interest in plasmonics, the results are novel, and the paper is well written. I therefore recommend accepting this paper for publication in Nature Communications after the authors address the following points:

- 1) The results presented in the paper are theoretical only and no experimental results are included. In addition, comparison to experimental results published elsewhere is very limited.
 - a) The authors should discuss in more detail how their theoretical results compare to previously reported experimental measurements.
 - b) If no such measurements exist, the authors should briefly discuss experiments which could validate their theoretical predictions.

- 2) "evidently, at the smallest separations and at large transitions orders n , our mesoscopic framework is pushed beyond its range of validity":
 - a) The authors should discuss in more detail the range of validity of their framework. In which regime is this framework no longer valid?
 - b) How can we infer from the results presented in the paper that the framework is not valid?
 - c) Is it possible to expand the range of validity of the framework?

Reviewer #4:

Remarks to the Author:

This paper presents intriguing and thought-provoking results on quantum effects in plasmon-emitter interactions. They are based on augmenting classical expressions for Mie coefficients and the like with Feibelman "d-parameters", the latter being calculated from some more explicitly quantum model for the metal (the authors use TDDFT jellium). I would say this paper does meet the criterion for publication in Nature Communications – it represents an important advance in the field of plasmonics/nano-optics. I have one minor comment:

Quantum effects are being added at a certain level of approximation – leading order in some sense. I believe the emitter is still a classical dipole. Others, e.g. Hughes and co-workers (PRB 85, 075303 (2012) and perhaps more recent work) take a different approach and treat the emitters as two-level systems and also quantize the electromagnetic fields. Quantum effects, perhaps of a different nature, then arise out of these types of approaches. Can the authors comment in the text on such approaches and the relation to their approach?

Reply to Referee 1

REPORT SUMMARY

In this paper, the authors use a theoretical treatment of mesoscopic electrodynamics based on Feibelman d -parameters to investigate the effect of nonclassical effects on a broad array of plasmon–emitter interactions, including dipolar and multipolar spontaneous emission enhancement, plasmon-assisted energy transfer, and enhancement of two-photon transitions. This topic is of broad interest in plasmonics, the results are novel, and the paper is well written. I therefore recommend accepting this paper for publication in Nature Communications after the authors address the following points.

REPLY SUMMARY

We are grateful to the referee for their positive report recommending our manuscript for publication and also for their thorough and thoughtful review of our manuscript.

Specific comments and questions

COMMENT 1.1

The results presented in the paper are theoretical only and no experimental results are included. In addition, comparison to experimental results published elsewhere is very limited.

- (a) The authors should discuss in more detail how their theoretical results compare to previously reported experimental measurements.
- (b) If no such measurements exist, the authors should briefly discuss experiments which could validate their theoretical predictions.

REPLY 1.1

We thank the referee for bringing up this crucial point, which our original manuscript had perhaps not elaborated sufficiently. Indeed, previously reported experimental work on plasmonic Purcell enhancement (e.g., Refs. R1–R2) already suggest the emergence of non-negligible nonclassical deviations for emitter–metal separations below 5–10 nm. We contend that the theoretical framework and predictions presented in our manuscript could explain these deviations; of course, a quantitative comparison would require a more elaborate and detailed analysis which is beyond the scope of our present work.^{1*}

More generally, we note that nonclassical effects in plasmon-enabled light–matter interactions may have been overlooked in past experimental studies: the deviations, though sizable at the smallest separations, can be small at intermediate separations. Such oversight could occur straightforwardly, if e.g. measured rates are fitted to an unknown experimental quantity—e.g. (environmental or plasmonic) bulk permittivity or emitter–surface separation—over a broad parameter region (rather than measured independently).

We emphasize that the emitter–surface separations considered in this work are well-within current experimental capabilities, as exemplified, for instance, by Refs. R4 and R5 (separations on the order of 1 nm). Parenthetically, we note that the quest for plasmon-enabled strong coupling—and the concomitant pursuit of ever smaller emitter–surface separations—will require new theoretical frameworks to guide and push experiments beyond the validity of classical, local-response theory; our results suggest a practical and elegant approach to achieve this.

Spurred by the referee’s comment, we now discuss in our revised manuscript how the

prediction of nonclassical corrections in light–matter interactions could be tested and validated using existing experimental platforms, and also further emphasize that the considered emitter–surface separations are well-within current experimental capabilities.

^{1*}A quantitative comparison would, e.g., require a precise and independent measurement of all classical *bulk* response functions, in order to ensure an accurate classical description and reliable “baseline” (as done in Ref. R3). This may be particularly important for the thin polymer or oxide layers that emitters are typically embedded in.

COMMENT 1.2

“[. . .] evidently, at the smallest separations and at large transitions orders n , our mesoscopic framework is pushed beyond its range of validity [. . .]”:

- (a) The authors should discuss in more detail the range of validity of their framework. In which regime is this framework no longer valid?
- (b) How can we infer from the results presented in the paper that the framework is not valid?
- (c) Is it possible to expand the range of validity of the framework?

REPLY 1.2

As the referee rightly points out, the d -parameter framework^{2*} and our treatment of the emitter invariably incorporates a set of assumptions—and the overall treatment’s range of validity and applicability is necessarily bounded by those assumptions. Their net sum is a rough boundary of applicability to feature sizes $\gtrsim 1$ nm (e.g., separation, radii, etc.); i.e. well-below the parameter ranges that we considered. Spurred by the referee’s comment, we now summarize the core assumptions underlying the d -parameter framework and our treatment of the emitter in the Supplementary Information (SI), in the newly added Supplementary Section S10. Below, we include a discussion derived from these new additions:

RANGE OF VALIDITY

The assumptions underlying the d -parameter framework are:

Dipole expansion: The d -parameters emerge from an interface-centered multipole expansion of the quantum mechanical charge and current density. The monopole term gives the classical framework; the dipole term produces d_{\perp} and d_{\parallel} ; the general n th order multipole term is of order $\sim (k_{\text{eff}}x)^n$, with x a length scale (e.g. d_{\perp} and d_{\parallel} at $n = 1$) and k_{eff} an effective modal wavevector (e.g., for a planar interface, the plasmon momentum k ; for the dipole resonance of a sphere, the inverse radius R^{-1}). Truncation at the dipole term thus produces a leading-order formalism, i.e. we require that $\{|kd_{\perp, \parallel}|, |R^{-1}d_{\perp, \parallel}|\} \ll 1$. Since $|d_{\perp, \parallel}|$ is Ångström-scale, this is generally very well-satisfied.

Local curvature: The use of d -parameters—whose properties derive from *planar* interfaces—at curved interfaces implies an assumption of “local flatness”. For a sphere, this is equivalent to requiring $R \gg |d_{\perp, \parallel}|$, i.e. imposes no additional restrictions relative to the dipole-expansion itself.

d -parameter nonlocality: The d -parameters are themselves k -dependent, i.e. nonlocal. This nonlocality only produces corrections of order $O(k_{\text{eff}}^3)$, cf. the overall $O(k_{\text{eff}}d_{\perp, \parallel})$ scaling of d -parameter terms and the expansion $d_{\perp, \parallel}(k) = d_{\perp, \parallel}(0) + \frac{1}{2}d''_{\perp, \parallel}(0)k_{\text{eff}}^2 + \dots$. The impact of intrinsic d -parameter nonlocality is consequently comparable to the omitted quadrupole and octupole surface terms (i.e. negligible).

Surface-centric: As the d -parameters are *surface*-response quantities they cannot describe quantum-size effects [R6, R7] (fragmentation of the electronic band structure into discrete levels) nor quantized volume plasmons (the splitting of the bulk plasmon dispersion $\omega_p(k)$ into a set of discrete “levels”) that arise in finite structures.^{2†}

The assumptions underlying our treatment of the emitter are:

Point-emitter: We use macroscopic QED to couple the states of the emitter and the electromagnetic modes of the plasmonic object (Supplementary Section S8). While macroscopic QED does not require a point-emitter approximation, we have adopted one^{2‡} in our calculations. As a result, emitter-size effects are neglected. While these effects are interesting—and can be substantial in large emitters such as quantum dots or molecules—we have omitted them to unambiguously identify the impact of mesoscopic corrections from the metallic surface-response alone (see also Reply 2.1).

Surface hybridization: We neglect wave-function overlap between the electronic states of the emitter and metallic surface; the inclusion of which would materialize as a (complex) self-energy renormalization of the emitter’s energy levels. Such a renormalization is negligible for the separations considered in our manuscript, where wave-function overlap is always vanishingly small. Of course, if the emitter is adsorbed to the metal surface—i.e. resides within its spill-out region—this could be a sizable effect. Similarly, we ignore image-charge effects on the emitter’s electronic orbitals. In sum, we treat the emitter’s intrinsic electronic structure as independent from the metallic surface.

Note that we do not generally constrain our considerations to two-level emitters; e.g., our consideration of multipolar transitions and two-photon emission feature the hydrogen levels. Naturally, all considered processes *could* be mapped to an equivalent two-level system (except two-photon emission which includes virtual transitions between *all* levels).

BREAKDOWN OF FRAMEWORK

The sum of the above-noted assumptions translates, broadly speaking, to the restrictions that $\{|k_{\text{eff}}d_{\perp, \parallel}|, |k_{\text{eff}}l_{\text{orb}}|\} \ll 1$, with l_{orb} denoting a characteristic size of the emitter orbitals. The effective wavevector of a given transition depends on the transition type and the emitter–surface separation h : e.g., for an n th order multipole transition, it is $k_{\text{eff}} \sim n/h$ [corresponding to the dominant wavevector in Eq. (8); equivalently, the maximum of the geometric factor $k^{2n}e^{-2kh}$]. As a result of this scaling with n , high-order multipole transitions will enter a regime beyond the framework’s validity sooner than e.g. the dipole transition. Figure 4b,f of our original manuscript highlighted this by a dotted line; in our revised manuscript, we have made efforts to clarify this further.

EXTENSIONS

Our framework could be extended beyond its present range of validity by lifting the approximations noted above, e.g. by including the quadrupole term in the d -parameter framework, or by incorporating emitter-size effects (see Comment 2.1).

^{2*}We have separately elaborated in substantial detail the underlying assumptions of the d -parameter framework in the as-yet unpublished SI of Ref. R3; to avoid duplication, the added discussion of the d -parameter framework’s assumptions here is, by comparison, abridged.

^{2†}Though irrelevant in the single-interface systems that we consider, the d -parameter framework naturally cannot account for tunneling [R8] which is intrinsically a two-interface effect.

^{2‡}Obviously, for the $n > 1$ multipolar transitions, we cannot (and do not) ignore the spatial extent of the transition orbitals. Even there, however, we make an approximation that is essentially analogous to

the point-emitter approximation familiar from dipole-transitions: the n th derivative of the field is assumed constant across the emitter's extent [see Eq. (S60)]. For the hydrogen-like orbitals that we consider, this is an exceedingly good approximation [R9].

Reply to Referee 2

REPORT SUMMARY

This paper presents intriguing and thought-provoking results on quantum effects in plasmon–emitter interactions. They are based on augmenting classical expressions for Mie coefficients and the like with Feibelman “ d -parameters”, the latter being calculated from some more explicitly quantum model for the metal (the authors use TDDFT jellium). I would say this paper does meet the criterion for publication in Nature Communications – it represents an important advance in the field of plasmonics/nano-optics.

REPLY SUMMARY

We thank the referee for their clear and judicious review and for their positive assessment of our work.

Specific comments and questions

COMMENT 2.1

I have one minor comment: Quantum effects are being added at a certain level of approximation—leading order in some sense. I believe the emitter is still a classical dipole. Others, e.g. Hughes and co-workers (PRB 85, 075303 (2012) and perhaps more recent work) take a different approach and treat the emitters as two-level systems and also quantize the electromagnetic fields. Quantum effects, perhaps of a different nature, then arise out of these types of approaches. Can the authors comment in the text on such approaches and the relation to their approach?

REPLY 2.1

We thank the referee for this comment. In fact, our theoretical formalism *does* treat the emitter as a quantum mechanical system, with two or, indeed, multiple levels. We use macroscopic quantum electrodynamics (QED), see Supplementary Section S8, to quantize the electric field using the dyadic Green function $\vec{\mathbf{G}}$ (expanded in the *mesoscopic* scattering coefficients), in the same spirit as in the noted reference by Van Vlack *et al.* [R10].

While Van Vlack *et al.* [R10] focus on the so-called non-dipolar corrections (a quantum effect derived from the finite extent of the emitter’s orbitals) to the Purcell factor for the dipole-allowed transition, we have focused on the impact on light–matter interactions from quantum effects in the response of the plasmonic metal. As the referee notes, these approaches are indeed different, examining complementary aspects of quantum corrections: one with origins in the emitter, the other in the plasmonic metal.

Expanding on this, we note that the approximation in our treatment of the emitter is not whether it is a classical or quantum emitter per se (they produce identical transition enhancement factors, i.e. when normalizing by the corresponding free-space decay rate [R11, R12]), but rather the assumption of a point-like emitter. For instance, if the finite size of the emitter’s orbital is included in the calculation of the dipolar Purcell enhancement, the computation involves the integration of $\hat{\mathbf{n}} \cdot \text{Im} \vec{\mathbf{G}}(\mathbf{r}, \mathbf{r}') \cdot \hat{\mathbf{n}}$ over a nonlocal interaction kernel that depends on the specific transition and emitter [R10, R13–R16]; this is of course exactly what Van Vlack *et al.* [R10] investigated. We emphasize that our framework can be straightforwardly adapted to incorporate these effects, simply by omitting the point-like approximation introduced in Eq. (S60), proceeding instead directly from Eq. (S53).

We thank the referee again for raising this important and pertinent point. In our revised manuscript, we have included additional discussion of this separate class of emitter-derived quantum corrections in both the main text and the Supplementary Material.

List of changes

Changes in the manuscript and Supplementary Information (SI) are highlighted by color (new or revised sentences in green; removed sentences in red).

Arising from referee comments

Spurred by the referee's comments, we have made a number of changes to the manuscript and SI, as detailed on the preceding pages. We include a summary below:

- Additional discussion of opportunities for experimental validation and existing supporting experimental results in Discussion section. \mapsto Comment 1.1
- New Supplementary Section S10 in the SI on framework assumptions. \mapsto Comment 1.2
- Added discussion of emitter-centric quantum corrections in Discussion section and Supplementary Section S8 of the SI. \mapsto Comment 2.1

Other minor edits

In addition, we note a few minor edits and a single inconsequential error-correction:

- Restated Eqs. (1) in terms of the “propagating” wavevector $k_{z,j} = (k_j^2 - q^2)^{1/2} = i\kappa_j$ instead of the “decaying” equivalent $\kappa_j = (q^2 - k_j^2)^{1/2}$ (where $j \in \{d,m\}$). This is simply an equivalent but clearer way of writing the reflection coefficients.
- Consolidated a mixed terminology of ‘quasi-static’, ‘electrostatic’, and ‘nonretarded’ for the $c \rightarrow \infty$ limit to a single variant: ‘nonretarded’.
- Corrected a trivial scaling error (a missing multiplicative constant ≈ 2.7) in Figs. 6b & 6e.

Changes requested by editorial office

EDITORIAL REQUEST 1

| All Nature Communications manuscripts must include a section titled “Data Availability” as a separate section after the Methods section but before the References

CHANGE 1

| We have included a Data Availability section in our revised manuscript.

EDITORIAL REQUEST 2

| Editorial policy checklist.

CHANGE 2

| The editorial policy checklist is completed and uploaded along with our revised manuscript and reply.

References

- [R1] K. J. Russell, T.-L. Liu, S. Cui, and E. L. Hu, *Nat. Photonics* **6**, 459 (2012).
- [R2] P. Anger, P. Bharadwaj, and L. Novotny, *Phys. Rev. Lett.* **96**, 113002 (2006).
- [R3] Y. Yang, Z. Di, W. Yan, A. Agarwal, M. Zheng, J. D. Joannopoulos, P. Lalanne, T. Christensen, K. K. Berggren, and M. Soljačić, [arXiv:1901.03988](https://arxiv.org/abs/1901.03988) (2019).
- [R4] R. Chikkaraddy, B. de Nijs, F. Benz, S. J. Barrow, O. A. Scherman, E. Rosta, A. Demetriadou, P. Fox, O. Hess, and J. J. Baumberg, *Nature* **535**, 127 (2016).
- [R5] F. Benz, M. K. Schmidt, A. Dreismann, R. Chikkaraddy, Y. Zhang, A. Demetriadou, C. Carnegie, H. Ohadi, B. de Nijs, R. Esteban, J. Aizpurua, and J. J. Baumberg, *Science* **354**, 726 (2016).
- [R6] W.P. Halperin, *Rev. Mod. Phys.* **58**, 533 (1986).
- [R7] R. C. Monreal, T. J. Antosiewicz, and S. P. Apell, *New J. Phys.* **15**, 083044 (2013).
- [R8] R. Esteban, A.G. Borisov, P. Nordlander, and J. Aizpurua, *Nat. Commun.* **3**, 825 (2012).
- [R9] N. Rivera, I. Kaminer, B. Zhen, J. D. Joannopoulos, and M. Soljačić, *Science* **353**, 263 (2016).
- [R10] C. Van Vlack, P. T. Kristensen, and S. Hughes, *Phys. Rev. B* **85**, 075303 (2012).
- [R11] L. Novotny and B. Hecht, *Principles of Nano-Optics*, 2nd ed. (Cambridge University Press, 2012).
- [R12] V. V. Klimov and M. Ducloy, *Phys. Rev. A* **72**, 043809 (2005).
- [R13] P. Tighineanu, A. S. Sørensen, S. Stobbe, and P. Lodahl, in *Quantum Dots for Quantum Information Technologies* (Springer, 2017) pp. 165–198.
- [R14] S. Stobbe, P. T. Kristensen, J. E. Mortensen, J. M. Hvam, J. Mørk, and P. Lodahl, *Phys. Rev. B* **86**, 085304 (2012).
- [R15] K. Jun Ahn and A. Knorr, *Phys. Rev. B* **68**, 161307 (2003).
- [R16] M. L. Andersen, S. Stobbe, A. S. Sørensen, and P. Lodahl, *Nat. Phys.* **7**, 215 (2011).

Reviewers' Comments:

Reviewer #1:

Remarks to the Author:

The authors thoroughly addressed all questions in my review. I recommend publication of the paper in its current form.

Reviewer #2:

Remarks to the Author:

The authors have very thoroughly addressed both reviewers' comments. The manuscript is considerably clearer and more impactful. It is now acceptable for publication in Nature Communications.